# Could fish aggregation at ocean aquaculture augment wild populations and local fisheries?

**Jessica L. Couture**[1]*, **Darcy Bradley**[1,2], **Benjamin S. Halpern**[1,3], **Steven D. Gaines**[1]

**1** Bren School of Environmental Science and Management, University of California, Santa Barbara, California, United States of America, **2** Marine Science Institute, University of California, Santa Barbara, California, United States of America, **3** National Center for Ecological Analysis and Synthesis, University of California, Santa Barbara, California, United States of America

\* jcouture@ucsb.edu

**Data Availability Statement:** Data analyses for this work are publicly available here: github.com/couture322/oceanFarms.

**Funding:** This work was supported by the National Oceanic and Atmospheric Administration Sea Grant

## Abstract

The global population consumes more seafood from aquaculture today than from capture fisheries and although the aquaculture industry continues to grow, both seafood sectors will continue to be important to the global food supply into the future. As farming continues to expand into ocean systems, understanding how wild populations and fisheries will interact with farms will be increasingly important to informing sustainable ocean planning and management. Using a spatially explicit population and fishing model we simulate several impacts from ocean aquaculture (i.e., aggregation, protection from fishing, and impacts on fitness) to evaluate the mechanisms underlying interactions between aquaculture, wild populations and fisheries. We find that aggregation of species to farms can increase the benefits of protection from fishing that a farm provides and can have greater impacts on more mobile species. Splitting total farm area into smaller farms can benefit fishery catches, whereas larger farms can provide greater ecological benefits through conservation of wild populations. Our results provide clear lessons on how to design and co-manage expanding ocean aquaculture along with wild capture ecosystem management to benefit fisheries or conservation objectives.

## 1. Introduction

Marine systems are increasingly impacted by growing demands on ocean resources, such that strategic management of expanding and emerging industries will be essential to support healthy ecosystems for biodiversity and continued uses. Despite improvements in fishery management, wild seafood catch has remained largely stagnant for several decades [1–3], and the potential upsides from reforms are modest [4] relative to the large projected global growth in demand in coming decades. Meanwhile, aquaculture is the fastest growing food sector in the world, and ocean spaces are considered the next frontier for aquaculture expansion [5, 6]. As the aquaculture industry further develops in marine environments, simultaneously maintaining the health and sustainability of wild populations and fisheries is critical. Such an approach is essential both as a backstop for biodiversity and for food security and a livelihood source for millions of people around the world [7]. Given the social and economic importance of wild fisheries and key ecological roles of targeted wild populations, understanding how potentially

program project (NA19OAR4170318) (JC) and the H. William Kuni Bren Research Award (JC). The funders had no role in study design, data collection and analysis, decision to publish, or preparation of the manuscript.

**Competing interests:** The authors have declared that no competing interests exist.

competing seafood production methods interact with fisheries will be important for optimizing co-management of these closely connected sectors and guiding sustainable seafood production into the future [8].

Marine aquaculture can alter the population dynamics of local wild species through three key mechanisms, which vary in their expected influence and intensity by farm type, farmed species, environment, and wild species characteristics, among other factors [9–11]. First, the added physical structure of the farm may aggregate wild species by providing structure for habitat. Case studies have documented attraction of wild organisms to farm areas [12–14] and increased biomass at and around farms [13, 15, 16]. Second, farming operations often restrict other ocean uses, including wild capture fisheries, which may impact the population dynamics of target species and the economics of the fisheries through the redistribution of fishing effort following farm installation [17–19]. Size, spacing, design, and other regulations will influence access to fish at and around farms. Third, the application and accumulation of excess feed in finfish farms, fouling on aquaculture infrastructure, and wastes from cultured organisms can also have positive (e.g., nutritional supplements) and negative (e.g., disease) impacts on the surrounding ecosystem [20–22]. Food subsidies, if moderate in scope, can provide a benefit to local wild populations, while excessive waste and parasite or disease transmission can cause harm (e.g., through alterations of natural mortality and growth rates) to wild populations [17, 18, 23–26] (Fig 1).

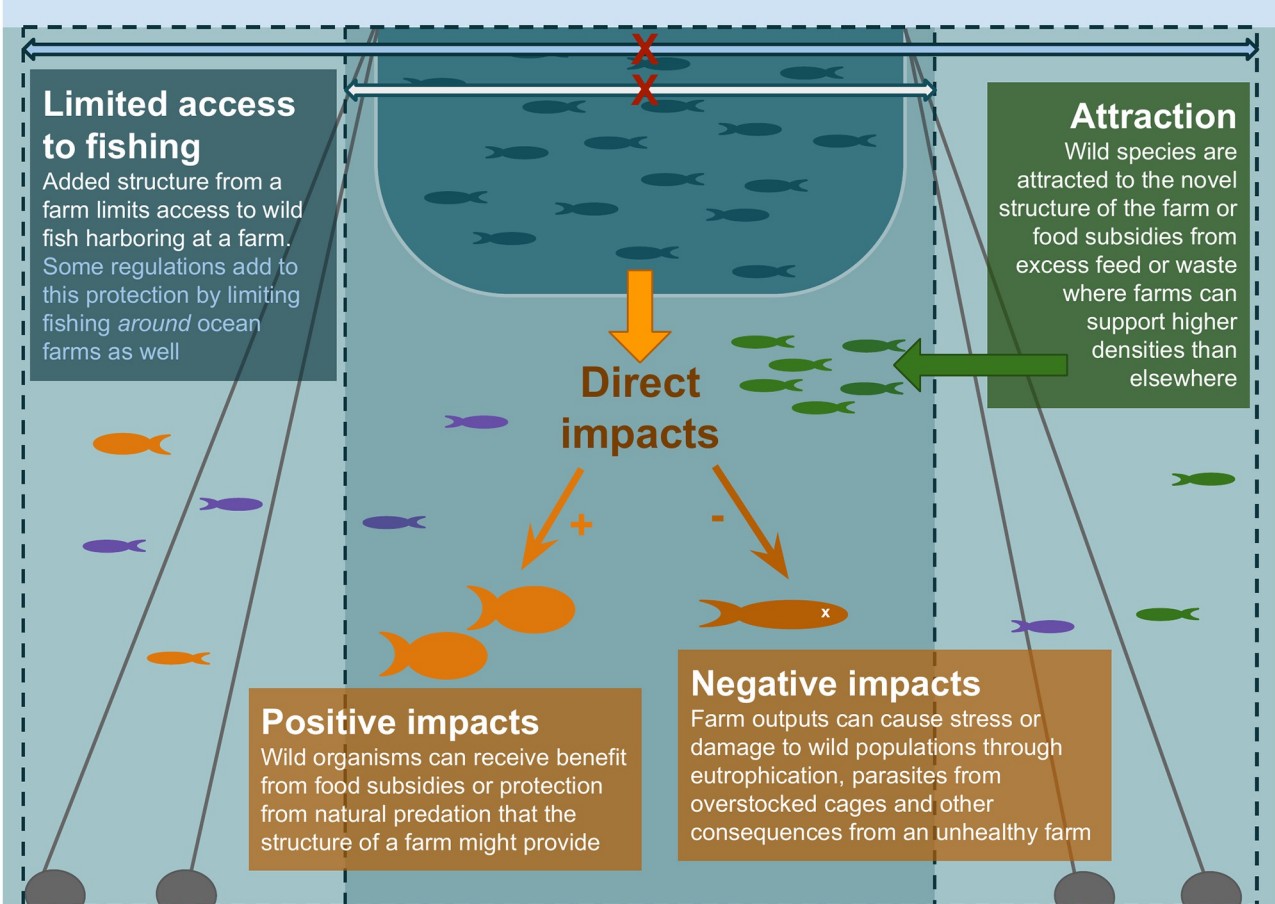

**Fig 1. Conceptual diagram of marine aquaculture interactions with wild populations and capture fisheries.**

Through these different mechanisms, marine aquaculture can act both as something akin to a fish aggregating device (FAD), with fish attracted to and accumulating around its structure [19, 27, 28], and as a mini-marine protected area (MPA). Farms effectively provide refuge from fishing even though they individually tend to be small ($<1$ km$^2$) [29] compared to MPAs (median size of 2.5 km$^2$, and can reach $>100{,}000$ km$^2$) [30]. Just as FADs are man-made structures used to concentrate naturally dispersed fish populations to increase fishing efficiency, aquaculture installations may provide a similar service to capture fisheries despite prohibiting fishing inside the farm itself, because fishers can fish the edge of the farm. FADs are especially effective for fisheries targeting pelagic species that tend to be highly mobile and broadly distributed [31, 32]. The growth of offshore ocean farms will likely increase interactions with pelagic fishes in ways that may make them more predictably accessible.

When fish become resident at farms [16, 33] the protection conferred by aquaculture farms could also function as an 'MPA', serving a stock rebuilding function, particularly in overfished or weakly regulated fisheries [19, 34, 35]. Such protections could potentially provide spillover benefits to surrounding fisheries for some stocks [36, 37]. For example, a seabream farm in the NW Mediterranean found that farms were conferring protection to commercially targeted fish stocks; however, harvest by farm employees inside the farm prevented stock rebuilding and spillover from occurring [19]. Beyond protection, farms could serve to further modify wild fish population dynamics through the addition of nutrient input, e.g., via excess feeds, to the surrounding environment. Depending on the context and quantity of inputs, such added nutrients may augment or undermine these purported population and fishery benefits. Ocean-based aquaculture research has thus far focused on optimizing production and minimizing negative impacts to ocean ecosystems with few emerging studies investigating potential social [38] and ecological [11, 39] benefits. Our work examines the mechanisms of these interactions to better inform how to manage coastal systems.

To explore the responses of wild populations and capture fisheries to marine aquaculture, we built a spatially explicit age-structured population and fishing dynamics model to simulate movement of wild fish and fishers around ocean farms. Our model tracks total biomass and catch biomass in the presence of farms relative to simulations with no farms as business as usual (BAU): $\delta B = \frac{B_{farm}}{B_{BAU}}$, where $B$ is total biomass of the simulated area, and $\delta C = \frac{C_{farm}}{C_{BAU}}$, where $C$ is catch biomass. The farm model simulates farm effects on wild populations by 1) aggregating fish in the farm area, 2) eliminating fishing within the farm, and 3) altering growth, reproduction and mortality rates within the farm (Fig 1). Farm effects are applied to a population dynamics and fishing model, which simulates movement, growth, reproduction, and mortality (natural and fishing) of wild fish and fishery yields [40]. As farms are added, we modify fish movement by varying the rate at which fish aggregate around farm infrastructure through an increased likelihood of movement toward a farm (attraction), in addition to the random density dependent movement that is simulated in the BAU model. Individual growth rates and natural mortality rates are altered at the farm (increased/decreased) to assess potential population consequences of residence at the farm with altered conditions [10]. Farms are tested as both a single large farm area and multiple smaller farms of uniform size to understand the effects of different planning and siting approaches on outcome variables.

We use the model to estimate changes to total biomass and fishery biomass (i.e., catch) within a region with a farm or network of farms relative to BAU (i.e., no farms present). We explore effects due to (*i*) aggregation at farms, (*ii*) farm design, and (*iii*) fitness impacts due to farms. We test each of these impacts across a realistic parameter space to understand the range of possible outcomes, and we vary multiple parameters at a time to investigate potential interactions. Each factor is additionally tested under strong (levels that achieve maximum

sustainable yield) and weak (open access) regional fishery management and across a wide range of total farm areas.

## 2. Methods

To simulate how ocean aquaculture farms might affect wild populations and wild capture fisheries we use the theoretical population and fishing model from Ovando et al. 2016 that focuses on potential MPA implications to fisheries. Briefly, the Ovando et al. 2016 model simulates fish movement and fishery behaviors around areas protected from fishing to test the biological and economic outcomes of different management scenarios. We then expand this model to include two additional features: 1) the potential attractive impacts of farms on fish dynamics similar to the roles of FADs, and 2) the potential negative/positive impacts of farms on individual fish performance (e.g., through disease, pollutants, or food augmentation). In particular, we adapt the model to include these farm-specific characteristics to explore how ocean aquaculture farms may affect population biomass and fishery catch. We implement the model using an annual time step in a simulation area that is large enough to capture changes in parameters of interest; population and fishery outcomes are reported at equilibrium as the difference between population and fishery outcomes with and without the addition of a farm. All modeling and analyses were conducted in R [41] and all data and code are available online: *github.com/couture322/oceanFarms*.

### 2.1 The business as usual (BAU) model

**2.1.1 Population model.**   The model is a single-species deterministic model that represents the range of a theoretical fishery stock divided into spatial patches [40]. All scenarios were run on a 1-dimensional closed system of 100 patches of identical size, with no immigration into or emigration out of the system. Patches are connected via larval dispersal and adult movement. Larval dispersal is uniform across all patches with density-dependent survival by patch to emphasize the impacts of adult movement and spillover on our results [42].

Adult movement is calculated for each time step using a Gaussian movement kernel to calculate the probability of moving from one patch to another. Probabilities are based on the distance between patches, and a static movement parameter that scales movement to specified species mobility. The probability of movement from patch $i$ to patch $j$ is

$$p_{i,j} = e^{\frac{-d_{i,j}^2}{2\sigma_m^2}}$$
(1)

where $d$ is the distance between $i$ and $j$ in number of patches and $\sigma_m$ scales movement for a species based on a species range parameter. The sum of movement probabilities from a given patch, $p_i$, to all other patches is 1. Edges are wrapped to avoid edge effects in movement.

Carrying capacity is calculated as the equilibrium stock biomass with no fishing and no farms and is applied uniformly across all patches in the BAU model. Fish length is calculated using the von Bertalanffy growth equation for each age group and adjusted based on the patch carrying capacity. Average weight-at-age, $w_{age}$, is thus density dependent and calculated as:

$$w_{age+1} = w_{age} + \left(b_2 L_{age+1}^{b_1} - b_2 L_{age}^{b_1}\right)\left(1 - \frac{bm_{patch}}{K_{patch}}\right)$$
(2)

where $b_1$ and $b_2$ are weight-at-age constants and $L$ is length. $bm_{patch}$ is the total biomass

occupying the patch at a given time step, and $K_{patch}$ is the carrying capacity for total biomass in a patch.

**2.1.2 Fishing dynamics.** Impacts of ocean farms on capture fisheries resources were considered under weak and strong fishery management regimes. Weak fishery management is represented by a high-value fishery under open access management. In this case, fishing effort is calculated and adjusted at each timestep at the level that maximizes profits. Strong fishery management is represented by a constant effort that generates maximum sustainable yield for the simulated stock. In both scenarios, total fishing effort is tracked rather than individual fishers or boats in order to simplify the fishing models. Fishers are assumed to be knowledgeable about where fish biomass is highest and so gravitate to more profitable fishing patches. Therefore, effort in a given patch is scaled based on fishable biomass available in a given patch. Catch biomass, $C_{i,t}$, is tracked by patch, $i$, and time-step, $t$, as

$$C_{i,t} = \sum_a \frac{F_{i,t} v_a}{F_{i,t} v_a + M} N_{i,a,t} (1 - e^{-M - F_{i,t} v_a}) w_a \qquad (3)$$

where $F$ is fishing mortality, $M$ is natural mortality, and $v_a$ is selectivity at age. Fishing mortality is removed from the total population biomass.

## 2.2 The aquaculture farms model

To test the impacts of ocean farms on wild populations and fisheries, we modified the Ovando et al. 2016 MPA model which addressed protection from fishing to incorporate two potential additional farm impacts: 1) aggregation modeled as attraction to farms because of altered habitat and increased carrying capacity at farms, 2) impacts on individual fitness (Fig 1). These three elements are varied across parameter spaces to assess how each affects population biomass and fishery catches. Farm design is also changed by varying the number and size of farms to understand the role of farm siting. Particular patches in the 100-patch array were indicated as farm patches based on the farm size and design scenario being run, and these farm patches experienced the modifications described below.

**2.2.1 Aggregation at farms.** Aggregation was modeled as attraction, $A$, to farm spaces because of increased individual growth rates at farms. Attraction to the farm controlled the rate at which individuals were able to aggregate and was modeled by an increased likelihood of directional movement towards the farm when within an indicated vicinity, or zone of influence (ZOI). Therefore, attraction, $A$, increased the likelihood of movement to farm patches, $j_{farm}$ from patches within a designated distance from the farm, $i_{ZOI}$, relative to movement to other patches, $j_{non-farm}$, where $A$ is a scalar multiplier greater than one:

$$p_{i_{ZOI}, j_{farm}} = e^{\frac{-d_{i,j}^2}{A 2 \sigma_m^2}} \qquad (4)$$

Since attraction is difficult to quantify, the attraction parameter was varied over a range of values from 1 (no attraction) to 15. Case studies report densities of up to 20 times that found away from farms [18, 43], which represents a combination of attraction and ability to retain individuals, so we used 15x attraction as a conservative upper limit. The ability to build up biomass at the farm was modeled with increased individual growth rates at farms which was controlled by increasing patch-level carrying capacity at a constant rate, $K_{farm} = 3 \times K_{patch}$. $K$ acts to limit growth of individuals as indicated in Eq 2. Thus, farms increase stock carrying capacity by allowing additional biomass at farms. Density dependent movement was also applied as in the BAU model to maintain baseline movement and allow for spillover benefits from the farms.

**2.2.2 Protection from fishing.** To represent physical and regulatory limitations to fishing at farms, fishing was prohibited at farm sites and effort was redistributed to non-farm fishable spaces. Redistribution of fishing, $F_{i,t}$, for a given patch, $i$, and time-step, $t$, was calculated as:

$$F_{i,t} = F_{i,BAU} \frac{1}{1 - P_{farm}} \bar{B}_{i,t} \tag{5}$$

where $F_{i,BAU}$ is the fishing mortality for the given patch before farms were introduced, $P_{farm}$ is the proportion of patches in the farm, and $\bar{B}_{i,t}$ is total fishable biomass (biomass of legal size in non-farm patches) [44].

Farm design impacts were tested by dividing the total farm area into individual farms of different sizes. For each total farm area explored (N patches with farms), farm designs were tested as a single farm of the entire N patches (*farm size = $N_{patches}$*) and farms each divided up into clusters of 5 patches (*farm size = 5*).

$$Number\ of\ farms = \frac{N_{patches}}{farm\ size} \tag{6}$$

Farm size is modeled relative to the population's range size and is varied from 0% to 40% to represent a range of possible scenarios, including smaller farms in restricted spaces or affecting species with smaller home ranges, to large industrial farms affecting species of differing mobilities.

**2.2.3 Impacts to survival.** Direct impacts from marine aquaculture on the fitness of individuals in wild populations are tested as decreases and increases to natural mortality, to represent benefits and damage to wild populations, respectively. Carrying capacities were held constant for all farm scenarios, so impacts to natural mortality act in addition to changes to individual growth rate from carrying capacity at farms. Positive impacts were represented as a 10% decrease in natural mortality compared to BAU, and negative impacts to the population were simulated as a 10% increase in natural mortality. The 10% change in mortality is arbitrary and was selected to be large enough to result in detectable population impacts, but not so large as to overwhelm the population dynamics and become uninterpretable.

## 3. Results

### 3.1 Aggregation at farms

Aggregation at farms was modeled as a combination of behavioral responses to the altered habitat structure created by farms: attraction, varying movement towards a farm, and retention, increasing the likelihood of individual persistence around farms. Aggregation improved benefits to total biomass across all scenarios, with faster aggregation (higher levels of attraction) leading to greater increases in wild populations (S1 File). Modest levels of aggregation at farms also caused increases in total catch for both strongly and weakly managed fisheries, with a few notable exceptions (Fig 3). Under strong fisheries management (i.e., those fished at MSY), benefits resulted when total farm areas were low to moderate (areas < 15–25% of the total area depending on the level of attraction), with significant fishing losses occurring with larger farm areas. Biomass also increased more rapidly as attraction increased under strong fisheries management (S1 File). Increases in the levels of attraction eventually led to losses to total catches (up to 20% losses depending on total farm area), because too many fish became inaccessible. However, for all levels of attraction, total biomass of the population benefitted, increasing by 50–250% due to the protection from fishing that farms provided.

Under weak fishery management, fisheries' benefits from farms were even greater, with yields increasing up to 33% over BAU scenarios with no farms. Increasing attraction led to the greatest benefits for smaller total farm area (up to 10% farms) for weakly managed fisheries. These benefits to fishing occur only after a period of significant initial fisheries losses since the benefits require time for populations in farms to grow and subsequently spill over to fished areas. Time to recovery was delayed and losses were greater with increasing attraction due to greater initial removal of biomass from fishable areas following farm establishment. Accordingly, benefits to total biomass accumulated more quickly with higher levels of attraction (S1 File), because more fish were immediately protected from fishing.

**3.1.1 Species movement.** To understand how results might change for different species, we varied average adult movement from sessile (movement = 0, eg., mussel fisheries) to highly mobile pelagic species (movement = 60% of the total simulated space). When the fishery was well managed, catches benefitted from the farms across farm areas when species were moderately and highly mobile. When there was low or no movement, fishery catches declined compared to no farm for all farm areas (Fig 2). Similarly, when fishery management was weak, benefits to catches were highest for moderate and highly mobile species across farm areas. When movement was low, benefits were also realized, although less so. For sessile species under weak management, fishery losses were felt for all farm sizes (Fig 2). Biomass increased

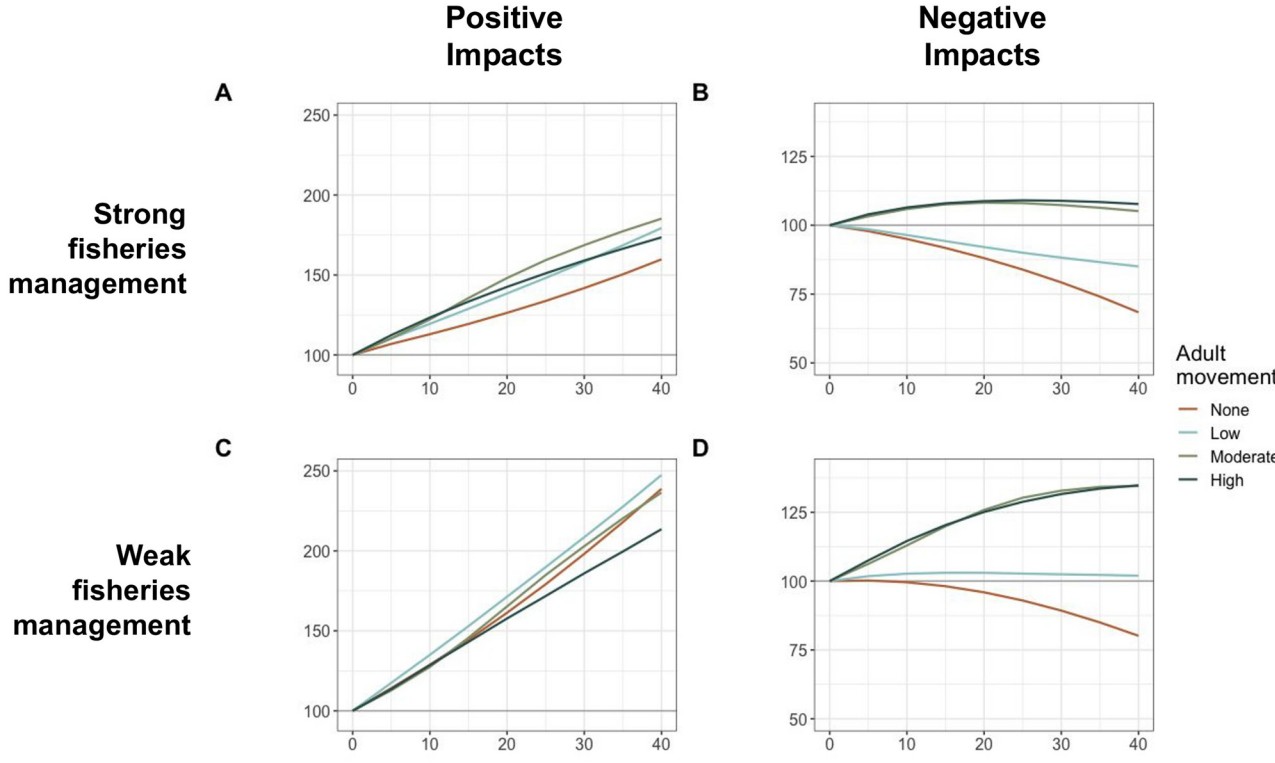

**Fig 2.** Equilibrium catch biomass (A, C) and fishery catches (B, D) relative to scenarios with no farm, given different movement rates over a range of total farm areas. Scenarios are tested under strong (A, B) and weak (C, D) fishery management. Farms are all divided into several smaller farms and here farms have no impact on wild population natural mortality rates.

for all movement rates under both strong and weak management, although to a greater degree under weak management (Fig 2 & S5 Fig in S1 File).

### 3.2 Farm design

Farm design was simulated in several configurations to test how siting and planning might affect the outcomes of farm impacts to wild capture fisheries. Total farm area was modeled as one large contiguous farm and as a network of equal sized smaller farms. Dividing up total farm area into networks of smaller farms increased the benefits of aggregation to fishing catches in both fishery management scenarios (Fig 3). Well managed fisheries experienced a maximum 9% increase in catches over BAU with multiple farms (compared to only a 4% maximum increase under the one large farm design). Similarly, open access fisheries saw a maximum increase in catches of 33% over BAU with multiple farms (compared to only an 11% maximum increase with one large farm). Smaller farms also shortened the time to recovery of fishing yields after farm introduction when management was weak. Initial fishing losses were less severe than with large contiguous farms (S1 File).

### 3.3 Impacts to survival from ocean farms

Effects of ocean farms on the survival of wild populations were simulated by modifying natural mortality for individuals that came into contact with farms. Negative impacts (e.g., from

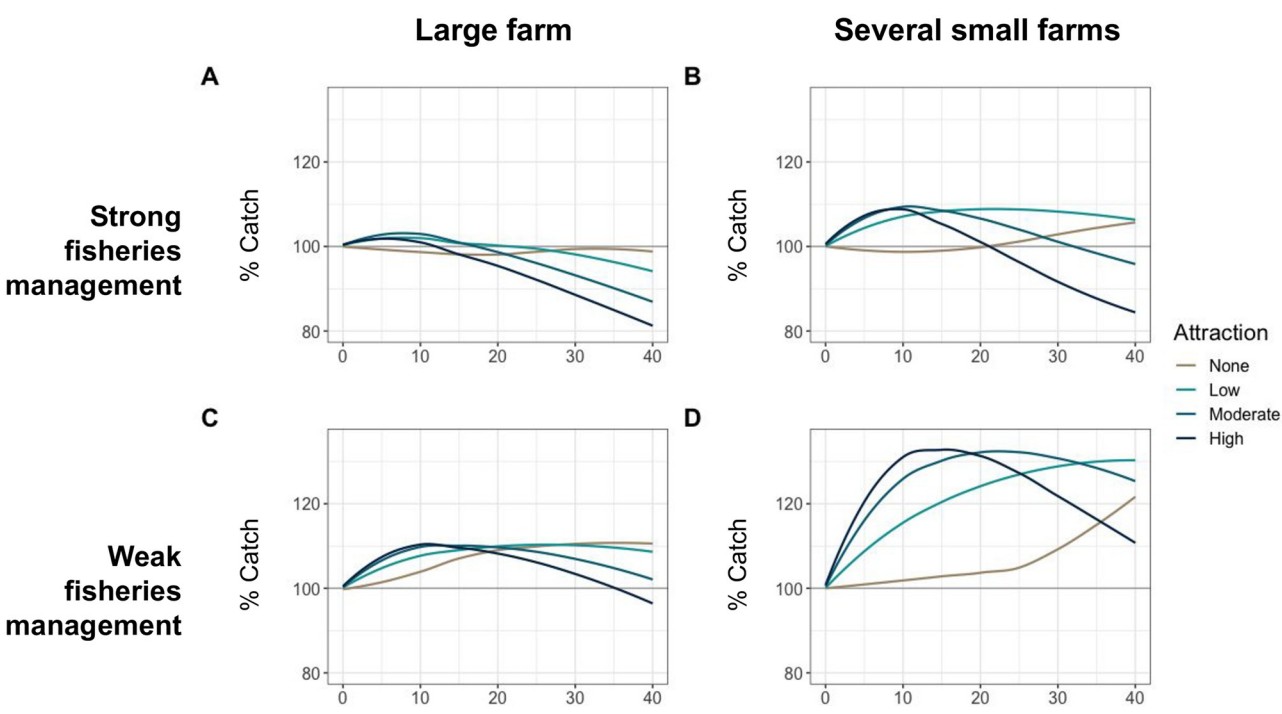

**Fig 3.** Equilibrium catch biomass relative to catches with no farm, given different levels of aggregation at farms, with farms of varying total area coverage modeled as one large contiguous farm (A, C) and the total farm area broken up into smaller separate farms (B, D). Top plots (A, B) are run under strong fisheries management, bottom plots (C, D) are under poor management.

pollution or disease) increased local natural mortality, and positive impacts (e.g., from food supplements) decreased local natural mortality. These impacts to the survival of wild organisms at aquaculture sites generally intensified or reduced benefits from farms across fishery management scenarios depending on the direction of mortality impacts (Fig 4). At high levels of damage, which cause significant mortality to wild populations (greater than fishing mortality), both total biomass and fishing yields saw substantial losses (S1 File). At low to moderate levels of harm to wild populations, however, farms were still able to provide benefits to both wild populations and fishing, especially for highly mobile species (Fig 4, S5 and S6 Figs in S1 File).

Farm derived benefits to wild populations coupled with attraction to the farm led to increases in catches for mobile species across farm areas (Fig 3A), although well managed sessile species saw fishery losses. Low mobility species maintained catches relatively consistent with pre-farm catches when benefits were derived from the farms. Negative impacts to wild populations at the farm caused declines in fishery catches for well managed fisheries. When fisheries were poorly managed, those targeting more mobile species did see benefits, despite damage to the wild populations (S1 File).

## 4. Discussion

The results here show that, in theory, ocean farms can have positive or negative impacts on wild populations and local capture fisheries depending on several design attributes and traits

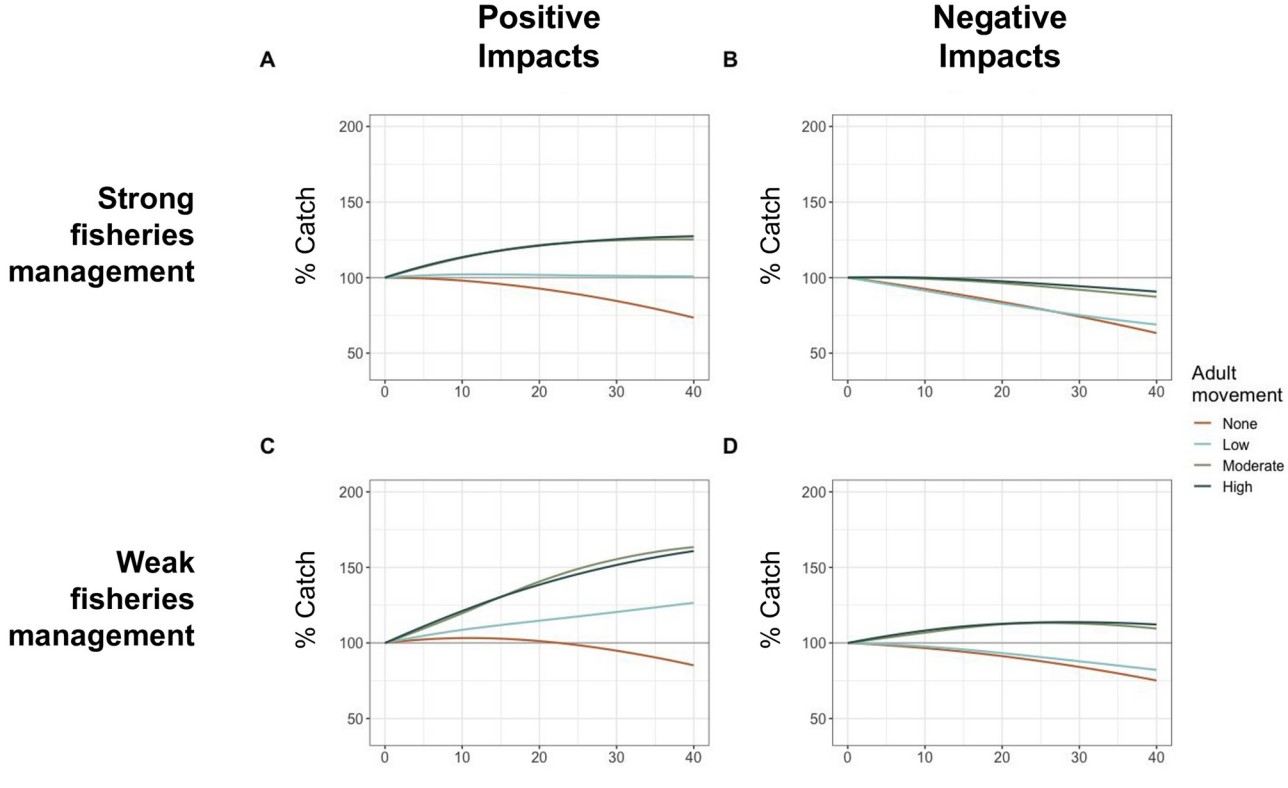

**Fig 4. Catch biomass relative to no farm over a range of total farm areas.** Farms are all divided into several smaller farms. Farm scenarios with positive (A, C) and negative (B, D) impacts to the wild population, under strong (A, B) and weak (C, D) fishery management.

of species of interest. For the vast majority of parameter combinations, results showed that farms could have positive conservation effects on overall fish biomass and disproportionate increases in total biomass compared to catches suggest that farms could provide a refuge for individuals that would act to replenish fished areas. The exception was only in the case of extreme negative impacts on fish mortality from farm, when mortality rates exceeded those from fishing. For effects of farms on fisheries yields, however, the direction and magnitude of impacts depended more critically on the mix of environmental and species influences. As movement slowed, benefits to growth and survival (from protection from fishing) were less efficiently distributed into fishable areas, with benefits to sessile species remaining inside the farm area. Fishery management was also important in determining the impacts of farms on catches, with potential benefits under both conditions, benefits were maximized under weak management. Smaller farms increased access to farms via an increased edge to area ratio, which increased aggregation rates of wild individuals more quickly, leading to more rapid benefits from increased aggregation sizes within farms and subsequent protection from fishing. In order to apply these insights to the strategic design and planning of an ocean farm, a clear understanding of each parameter is needed for a given ocean farm setting.

The degree of species' aggregation–including both attraction and retention in response to altered habitat quality—is difficult to measure empirically, but its strong potential role in driving benefits from farms to both fisheries catch and fish biomass warrant more focused empirical attention. Even as the marine aquaculture industry grows in many countries, total farm areas generally remain quite small compared to managed areas and movement capacities of many fished stocks, particularly as farms move offshore where species movement tends to be greater. Therefore, while we may see a potential for consistent benefits to total biomass from aggregation at a farm, as farms take up substantially more space these conservation benefits may ultimately come at a cost to fisheries yields if rates of aggregation are too high. Tradeoffs found between catch and biomass benefits in well managed fisheries are notable, although ocean farming across most of the globe still occurs at a scale small enough that benefits could be achieved from significant future expansions and increases in population biomass can help create a buffer to management uncertainty in the face of decreased catch. Where management is strong, requisite resources and local and institutional knowledge are more likely available to effectively adapt management to the new setting in a way that minimizes losses to fisheries. A better understanding of how fish behave around marine aquaculture farms will improve our ability to model interactions with marine farms and inform strategic planning of the design and scope of ocean farms to maximize benefits to both wild populations and local fishing.

Movement patterns around farms played a dominant role in our forecasts of outcomes for wild capture fisheries with the introduction of aquaculture. Unfortunately, our empirical understanding of behavioral responses of fishes, such as aggregation rates and their propensity to move from a farm across multiple fished species and farm types remains relatively limited, although the widespread use of FADs in many pelagic fisheries suggests that this attraction to new habitat likely plays a significant role in farm impacts [45, 46]. As ocean farms move further offshore [5, 47], the species they interact with will likely be even more highly mobile and migratory. As a result, farms will likely increasingly fill the FAD role of attracting fish to a known location without being able to provide much protection, which can enhance the predictability of high fishing yield locations, thereby enhancing potential economic yields even more than biomass yields. Fish associations with FADs are highly variable by species, season, other species found at the FAD [48]. Ocean farms could cause similar volatility in aggregations or more consistency or residency given regular biomass supplementation (feeds, wastes, and/or product mortality). Therefore, further empirical research is needed to characterize if and how ocean farms might cause interruptions to migrations and longer distance movement.

While data are lacking to properly quantify rates of aggregation at a farm, there is less information still on the retention times of wild species once they arrive at ocean farms, which is an important factor in quantifying the total aggregation effects of a farm.

Farms can have at least two direct conservation benefits. First, they add structural complexity to an area, which can provide habitat and other biodiversity benefits for many species [49, 50]. Second, our results show the potential for farms to provide strong benefits to conservation in an area by providing refuge from fishing under both weak and strong fisheries management. There may also be other conservation benefits, such as enhancing population size of key fish species [51, 52] and adding to their metapopulation structure. In order to strategically use farms as a conservation tool, though, we need a better understanding of how different species react to and interact with a farm is needed. Since farms introduce novel habitat, empirical work should also focus on the ecological implications of ocean farms including understanding how species will shift their habitat use and how changes in movement and habitat use will affect ecological relationships and ecosystem function.

Here we have modeled several species behaviors to mimic common conditions described at farms, but there are additional behaviors that require further exploration. One is the potential consequences of movement between farms. For example, *Pollachius virens* is an important fishery species with variable movement patterns–at times remaining relatively sedentary, but also embarking on longer migrations [53, 54]. These fish have been documented spending significant time at salmonid farms in Norway [16, 55], but also moving long distances between farms, thereby increasing connectivity between aquaculture sites and serving as potential vectors for parasites or diseases. This well-studied case of behavior change due to ocean farming demonstrates the need for further investigation into how ocean aquaculture uniquely affects different species. Lessons learned can then be used to better parameterize this model to further inform planning and management of regional aquaculture and to improve understanding of potential risks to cultured and wild species.

Interactions between aggregation at farms and smaller farm areas could provide synergistic benefits to local fishing, especially for smaller to moderate total farm areas (with catches increasing up to 35% over BAU). For farm areas >10% of the simulated area, benefits to fishing peaked at lower levels of attraction suggesting a stronger FAD effect from smaller farms. At low levels, aggregation increases the number of individuals receiving any benefits from the farm but is still weak enough to allow those that reaped the benefits from protected farm habitat to leave to fishable areas more frequently. Given the relatively limited amount of area currently occupied by marine aquaculture, these results suggest there is substantial scope for well-designed and managed farms to create future fisheries benefits. Increases in farm benefits to the growth of individual fish, which may be an important motivator for fish attraction, subsequently spilled over to predictably benefit nearby fishable areas. Smaller farm designs yielded benefits over BAU scenarios that were only achievable with far larger total farm areas in the absence of aggregation (Fig 3D). Given the potentially strong synergistic interactions between aggregation at farms and farm design, understanding the movement patterns of key stocks of interest and how such patterns will be modified by ocean aquaculture is important for strategically designing the ideal sizing and spacing of new ocean farms in different ecosystem settings.

Although our results suggest potential benefits from farm alterations to fish movement patterns and protection from fishing conferred by ocean farms, these benefits can be significantly compromised by detrimental farm effects on individual fish performance. Negative farm impacts need to be minimized to support a healthy ecosystem and a healthy fishery. Still, these results suggest that the expected trade-offs between different farm designs, modification to movement, and direct impacts to fitness can lead to surprising results, such as benefits to catches despite increased stress to individuals around farms (Fig 4D). Whether benefits or

losses to fishing would result from the introduction of a farm is contingent on these trade-offs between benefits from protection and damage from the farm. How these confounding impacts actually interact at farms has not been tested *in situ*, but we demonstrate here that complex outcomes from competing or conflicting impacts could lead to unanticipated results, highlighting the need for a better understanding of the interactions between these individual impacts and species of interest. Nonetheless, designing ocean farms to avoid such tradeoffs, wherever possible, are likely to provide greater conservation and fisheries benefits.

### 4.1 Real world applications

We show that the interaction between ocean farms and wild capture fisheries likely depend significantly on the design of the farm and the ecology of the target species. To connect our theoretical findings to practice, there is a need to parameterize our model with site- and species-specific information. Determining rates of aggregation at a given farm for a particular species is critically important but will be challenging. However, telemetry studies to track the movement of species of interest in and around farms can be used to provide an empirical measure of attraction. Similarly, to understand the extent to which carrying capacity might be altered at a farm, studies at existing farm sites on individual performance (growth and mortality) relative to non-farm areas would provide valuable insights. With such data, these model frameworks can be coupled with more commonly available parameters, such as fishery stock status and species-specific movement rates, to predict population and fishery responses from alternative farm designs.

High variability in movement rates and level of aggregation is common among species [10], and responses to farms can also vary temporally [56, 57]. Improved information on both of these factors would help refine our model and its predictions. Our focus here was to test how varying each model parameter might hypothetically affect outcomes. While ocean farms are small compared to species distributions, species with small home ranges could see large proportions of their experienced range occupied by a farm, with important consequences to local fisheries. Additionally, semi-enclosed systems such as lagoons, fjords or islets could quickly become dominated by aquaculture, analogous to the higher end of our tested total farm area range. Species with moderate to high movement rates might interact with offshore pelagic farms, whereas sessile and low movement species might be found at farms nearer to shore, including seaweed or oyster farms, particularly when considering species targeted for fishing. However, crabs and lobsters have relatively low movement rates and have been documented associating with deeper off-shore farms [27], and sessile species such as seaweeds and mussels recruit to novel farm structures in the photic zone regardless of how deep or far offshore the farm site might be. Thus, our model may be broadly relevant to many species in many contexts.

Often considered a nuisance to farmers, species that foul farm infrastructure can attract and feed predatory fish species [10]. Larger predatory fish are also attracted to ocean farms to prey on other aggregated wild species [58] or the biomass of the farmed product itself [59]. In many locations ocean farms create more of a dynamic system than was tested here. Multi-species and ecosystem level interactions at an ocean farm could change the outcomes found here and are important to understanding the real impacts of ocean farms on fishery and conservation goals.

We found that when used strategically, ocean-based aquaculture has the potential to benefit both wild populations and fishery catches. Empirical studies of local ecosystems, fisheries, management, and aquaculture goals will be essential to defining the context of each farm and informing strong coastal management. For example, understanding how movement dynamics might be further modified by ocean farms (e.g., which species will be more resident or transient

at farms? how are these processes affected by seasonality of farm use and migratory behavior?) will help optimize farm design and management in support of local objectives. FAD-like effects found for the "several small" farm design can enhance social sustainability by increasing fisheries catches, but there can be tradeoffs between fisheries and conservation objectives. If conservation is the priority, farms should instead be grouped together, or buffer zones that prohibit fishing around farms can be used to extend protections of aggregated individuals. To this end, managers are increasingly employing multi-use MPAs allowing aquaculture operations within no-take zones [38]. As we show here, aggregation at these farms may increase the protective benefits of the MPA to wild populations by attracting more fish to the MPA [19], as long as farms do not cause excessive harm to wild populations or ecosystems (S1 File). Both wild fisheries and marine aquaculture are predicted to be increasingly significant food sources for human populations globally [3, 60]; understanding how to best plan for and manage these coexisting industries will allow us to efficiently produce diverse seafood products while also supporting robust and sustainable coastal economies and ecosystems into the future.

## Supporting information

**S1 File. Supplemental S1–S6 Figs.**
(PDF)

## Acknowledgments

The authors would like to acknowledge the consultation and feedback provided by Dr. Dan Ovando, Dr. Reniel Cabral and Lennon Thomas throughout the modeling and analysis stages.

## Author Contributions

**Conceptualization:** Jessica L. Couture, Darcy Bradley, Benjamin S. Halpern, Steven D. Gaines.

**Data curation:** Jessica L. Couture.

**Formal analysis:** Jessica L. Couture.

**Funding acquisition:** Jessica L. Couture, Darcy Bradley, Steven D. Gaines.

**Investigation:** Jessica L. Couture.

**Methodology:** Jessica L. Couture, Darcy Bradley, Benjamin S. Halpern, Steven D. Gaines.

**Project administration:** Jessica L. Couture.

**Supervision:** Steven D. Gaines.

**Visualization:** Jessica L. Couture.

**Writing – original draft:** Jessica L. Couture.

**Writing – review & editing:** Jessica L. Couture, Darcy Bradley, Benjamin S. Halpern, Steven D. Gaines.

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
