## [Decision Letter · Decision Letter 0]

2 Oct 2023

PONE-D-23-28349Fish aggregation at ocean aquaculture can augment wild populations and local fisheriesPLOS ONE

Dear Dr. Couture,

Thank you for submitting your manuscript to PLOS ONE. After careful consideration, we feel that it has merit but does not fully meet PLOS ONE’s publication criteria as it currently stands. Therefore, we invite you to submit a revised version of the manuscript that addresses the points raised during the review process.

We look forward to receiving your revised manuscript.

Kind regards,

José A. Fernández Robledo, Ph.D.

Academic Editor

PLOS ONE

Journal Requirements:

3. We noted in your submission details that a portion of your manuscript may have been presented or published elsewhere:

"This work was included in the corresponding author's dissertation, but this version did not undergo peer review and results have changed significantly since the approval of the disseration and therefore, the lessons and discussion are also different. " 

**Additional Editor Comments:**

Dear Dr. Couture,

The peer reviews have been completed. Both reviewers commend your team for the extensive effort invested in model development. However, they express concerns regarding the practical applicability of the models in real-world scenarios and the attempt to encompass every conceivable situation and species within a single paper. Please address their concerns.

Sincerely,

-j

Reviewers' comments:

Reviewer's Responses to Questions

**Comments to the Author**

1. Is the manuscript technically sound, and do the data support the conclusions?

Reviewer #1: Yes

Reviewer #2: Partly

2. Has the statistical analysis been performed appropriately and rigorously? 

Reviewer #1: N/A

Reviewer #2: Yes

3. Have the authors made all data underlying the findings in their manuscript fully available?

Reviewer #1: Yes

Reviewer #2: Yes

4. Is the manuscript presented in an intelligible fashion and written in standard English?

Reviewer #1: No

Reviewer #2: Yes

5. Review Comments to the Author

Reviewer #1: The Authors successfully and clearly presented a theoretical model of biomass variation of wild resources in relation to offshore (local) farming systems at different regimes of management (with different patchiness), according to clear assumptions and considering adult, although simplified, "fish" behaviour.

Although I am not an expert of modelling, I appreciated the rationale at the basis of this research work, which could represent a winning solution for both biological conservation purposes and sustainment of coastal communities.

On the other hand, as stated by the Authors, the model should be successfully and effectively tested in the real world. In addition, the total lack of mention or consideration of specific traits of wild resources (e.g., species-specific seasonal migrations, reproductive traits and strategies, ecology and position into the food web) makes difficult the identification of real applications. If no specific case study has been developed yet, a focus on the ecological approach (keystone of an updated and rich bibliography related to responsible fishery) would be recommended. Indeed, references are a little outdated and the manuscript sounds of lacking pragmatism, as it is presented.

I recommend to boost the general practicality of the model proposed, keeping an eye on the ecological approach.

The English form is not always clean and clear in the descriptive sections; I recommend a deep revision of English soundness.

Lastly, here follow some minor comments.

INTRODUCTION

Line 58: please, modify "to supporting" into "supporting" or "to support".

Line 74: what the authors mean by "structure"? "Physical structures", "artificial structures for settlement"? Please, be more specific.

Line 75: please, delete "-" after "(12-14)" and add a space.

Line 78: "through the redistribution".

Lines 79-82: please, consider to modify the sentence as "Third, exceeding food provision in bony fish farms, fouling on aquaculture infrastructures, and waste from cultured organisms can also have positive (e.g., nutritional supplements) and negative (e.g., diseases) impacts on the surrounding ecosystem (20–22)".

Lines 82-83: please, modify "can benefit local wild populations" in "can represent a benefit for local wild populations".

Figure 1: please, improve the quality of the Figure proposed in terms of resolution.

Line 97: better "offshore ocean farms"?

METHODS

Lines 144-145: please, modify "The spatial scale was also theoretical and intentionally abstract to provide.." as "The spatial scale was also theoretically and intentionally abstracted to provide..".

Lines 145-146: please, change "All modeling and analysis was conducted in R(38) and all data and code are available here" in "Modeling and data analysis were conducted in R(38) and all data and code are available here".

Reviewer #2: Review of “Fish Aggregation at Ocean Aquaculture Can Augment Wild Populations and Local Fisheries”

The authors attempt to tackle the pressing issue of how an increasing aquaculture and fish farming industry may affect existing fisheries and wild target fish populations. The authors utilize robust modeling techniques to assess how catch and biomass would change based on various situations such as: large farms vs. many smaller farms, positive impacts via aggregating at farms vs. negative impacts via aggregating at farms, and various levels of both movement and attraction to the fads. The authors conclude that farms generally have a positive impact and can act as a refuge for fish to restock overfished populations. As a result, intelligent farm design can help increase catch for commercial fisheries and increase wild stocks. However, they state that more empirical information about fish behavior in regard to migration to, residency, and emigration away from fish farms as well as the direct impacts, whether net positive or negative, is needed for most fish species that may potentially interact with fish farms.

I would like to commend the authors for their submission as a lot of time and work went into the creation of these models, and a they show lot of attention to detail in how and what to account for within the model. However, I cannot recommend this paper for publication without major revisions. The authors are attempting to assess every and all situations/species within one paper, while also making strong conclusions based on completely simulated data. By assessing every species that may interact with fish farms, they render the model ecologically impossible and thus detract from their conclusions. For instance, pelagic fish do not aggregate at FADs long enough for them to provide anything close to a spillover effect on wild populations. Furthermore, the scale of farm coverage is ecologically impractical for many species. There are not many species in which you could cover 40% of their home ranges or occupied space with farms, whether those are spread out or a single farm (Fig 2). Because of this, the conclusions are more or less ecologically meaningless to most pelagic species. Lastly, as far as I can tell, there is no real data integrated into the model, whether it’s life history data or stock population data. Yet, the authors make strong conclusions on the impact on farms from completely simulated, abstract data. There is of course merit in modeling data-limited fields such as this one to assess possible outcomes, but the authors’ implications for the results of this paper extrapolated too far based on the real-world applicability of this study. In essence, this manuscript straddles the line between a well-thought-out, well-defined theoretical model, and a model with no validation that is only loosely applicable. It would suit the authors to choose which direction they would like to take the paper to streamline it.

I would suggest the authors make one of the following possible revisions to the paper. 1.) Vastly tone down any implications/conclusions from the manuscript and present the model as purely theoretical in its entirety for use in fisheries monitoring when data become more readily available in the future. This is the easiest, as the code is freely available at the time of reviewing this manuscript. 2.) Specify an ecological system with which you model the data or even a specific species. For example, only include pelagic species, or only include nearshore species. This will allow you to make the model more specific, and potentially more applicable. 3.) Validate this model with data from a fishery with known data to see how it performs with real data input as a case study for the model as presented.

The authors are of course, free to revise as they wish. However, this manuscript presents a model that is far too broad to make the statements found in the results and discussion. Below are specific points that should be addressed to increase the strength of the manuscript.

Line 63-67. This is a run-on, I would split this up.

Line 88-92. This is another run-on.

Line 88-98. Maybe I missed it, but is there any evidence that fish farms act as FADs? The only papers cited here don’t specify fish farms as FADs in any way. While I generally agree with you, attraction is a major variable within the model presented. I would add at least a couple citations backing up this statement.

Line 99-108. This is my biggest discrepancy with the authors. While yes, MPAs serve as stock rebuilding functions, is there any actual evidence that fish farms specifically act as MPAs? If they do act as MPAs, is there any evidence that they are effective? Besides the fact that farms physically restrict fishing, MPAs largely function because of strategic placement. They are used to protect a large breeding population, specific vulnerable life history stages, breeding grounds, entire ecosystems etc. Is the argument that a random fish farm in the middle of the ocean does the same thing? Does this only apply to fish farms that are specifically within MPAs already? Is the spillover argument that the fish in the farm are producing gametes that are fertilized by other farmed animals or the wild populations to increase wild populations? This is unclear and not very well supported within the text. Literature cited is related to marine reserves, but nothing is cited for fish farms acting as MPAs. While this is going into the weeds a little bit, this is a foundational assumption within the model. This needs to be well cited or have data provided very clearly otherwise it undermines the conclusions based on the model.

Line 135-140. Run-on sentence, split up. Also, it needs to be specified which values for the fish life history were used (growth, reproduction, etc), and I would put them here. If they were pulled from Ovando et al., 2016, specify which species. Again, weeds, but these are important variables.

Line 148. Very well-defined variables in this and the next section that make it clear what is being used in the model.

Line 150-153. This is a major point of contention within the model. After a brief look at Ovando et al.’s 2016 paper, they used a 1-dimensional model to represent fish dispersal around an island, correct? Is this a model framework that theoretically aligns with such a broad model as described in this manuscript? This model should make the same assumptions that the original paper did, and limit the conclusive statements provided to that situation, or alter the model framework to reflect a more general situation.

Line 172-174. These definitions make sense, but what are the actual inputs for the model, and how do they differ?

Line 200-202. Citation?

Line 206-207. This is why it is so important to provide evidence that fish farms are actively playing an MPA role and not just a FAD role. This is a major factor within the model that is largely unsupported.

Line 226-228. Why 10%? Is there any reason for this number or is 10% largely arbitrary?

Line 229. I think you should add all of the figures in the supplementary information to the main text. It’s better to show the results of the model instead of just talk about them.

Line 237-239. I think these are incredibly useful metrics being presented here. However, is even a 10% area realistic simply due to farming? These are farms that are ~1 square km, as cited in this manuscript. More evidence that the scope of this manuscript might need to be narrowed to make the results more believable.

Line 256. Sessile is not a life history pattern observed in fish. While this is a concise term within the model to indicate no movement out of the farm once settled, it does not describe fish ecology.

Line 297-303. I possibly missed it, but is there any graphical representation of the biomass under positive impacts? They are discussed in this paragraph but no figure is cited.

Line 323-340. I appreciate the author’s observations about the limitations of the model being presented. However, without providing any real-world data or evidence, why should the readers believe any of the results of this model can be applied to real-world ecosystems?

Line 347-350. Wait, isn’t the argument that fish farms reduce fishing pressure due to inaccessibility?

Line 350-352. There are plenty of FADs that exist at the moment. Is there any evidence that large FADs interrupt migration patterns right now?

Line 366. Have real species behaviors been included in the model? Or are these abstract representations of potential behaviors? A nitpick, but very important to the interpretation of results.

Line 375-376. I would remove this sentence or rephrase this. Would new data then be incorporated into the model presented, or should management be based solely on empirical data?

Line 410-412. This is why validation specifically from the authors is sorely needed. Presenting a model without validation reduces the usability of the model.

Line 417-420. This begs the question of why commonly available parameters weren’t included in the model in the first place. Wouldn’t that provide a stronger case for using this model?

Line 423-424. Case in point for why conclusions and conclusive language need to be softened within the manuscript.

Line 429-431. The range of species here all have different very different levels of movement, dispersal, age/length at maturation, and fecundity. Won’t these need to be modeled differently to account for the disparity in all of these variables?

Line 431-435. It’s not quite clear why these species are included here. None of these organisms, which again have very different life histories and dispersal methods, were included in the model. Won’t these species need a completely different model to get accurate results from simulated situations?

Line 455-457. Is there any evidence that this is true? Not all fish species are attracted to FADs.

Again, I would like to commend the authors on their contribution to the field. I think this is a timely, and important piece of work. However, there is too much of a discrepancy between the range of species abstractly included in the model and the actual ecology of these species.

6. PLOS authors have the option to publish the peer review history of their article (what does this mean?). If published, this will include your full peer review and any attached files.

Reviewer #1: No

Reviewer #2: No

---

## [Author Response · Author response to Decision Letter 0]

4 Dec 2023

The authors thank the editor and reviewers for their thoughtful feedback. We took considerable attention to the important points raised about the clarity of our messaging and applicability of the theoretical results presented here and have edited the text to clarify and build a stronger manuscript. We appreciate the acknowledgement of the time and effort we have put into this work. Please see below for more details about our responses. 

Specific responses

Reviewer 1

The Authors successfully and clearly presented a theoretical model of biomass variation of wild resources in relation to offshore (local) farming systems at different regimes of management (with different patchiness), according to clear assumptions and considering adult, although simplified, "fish" behaviour. Although I am not an expert of modelling, I appreciated the rationale at the basis of this research work, which could represent a winning solution for both biological conservation purposes and sustainment of coastal communities. On the other hand, as stated by the Authors, the model should be successfully and effectively tested in the real world. In addition, the total lack of mention or consideration of specific traits of wild resources (e.g., species-specific seasonal migrations, reproductive traits and strategies, ecology and position into the food web) makes difficult the identification of real applications. If no specific case study has been developed yet, a focus on the ecological approach (keystone of an updated and rich bibliography related to responsible fishery) would be recommended. Indeed, references are a little outdated and the manuscript sounds of lacking pragmatism, as it is presented. I recommend to boost the general practicality of the model proposed, keeping an eye on the ecological approach. The English form is not always clean and clear in the descriptive sections; I recommend a deep revision of English soundness.

INTRODUCTION

Line 58: please, modify "to supporting" into "supporting" or "to support". Edited

Line 74: what the authors mean by "structure"? "Physical structures", "artificial structures for settlement"? Please, be more specific. Edited for clarification to “…the added physical structure of the farm…”

Line 75: please, delete "-" after "(12-14)" and add a space. Edited

Line 78: "through the redistribution". Edited

Lines 79-82: please, consider to modify the sentence as "Third, exceeding food provision in bony fish farms, fouling on aquaculture infrastructures, and waste from cultured organisms can also have positive (e.g., nutritional supplements) and negative (e.g., diseases) impacts on the surrounding ecosystem (20–22)". The sentence has been edited for clarity to “Third, the application and accumulation of excess feed in finfish farms, …”

Lines 82-83: please, modify "can benefit local wild populations" in "can represent a benefit for local wild populations". Edited

Figure 1: please, improve the quality of the Figure proposed in terms of resolution. All main text figures have been resubmitted at higher resolutions

Line 97: better "offshore ocean farms"? Edited

METHODS

Lines 144-145: please, modify "The spatial scale was also theoretical and intentionally abstract to provide.." as "The spatial scale was also theoretically and intentionally abstracted to provide..". Edited

Lines 145-146: please, change "All modeling and analysis was conducted in R(38) and all data and code are available here" in "Modeling and data analysis were conducted in R(38) and all data and code are available here". Edited

Thank you for the feedback on the methods; we have rewritten the paragraph highlighted above as follows (and generally edited the methods section to be in present tense, as is the norm for simulation modeling):

“We then expand this model to include two additional features: 1) the potential attractive impacts of farms on fish dynamics similar to the roles of FADs, and 2) the potential negative/positive impacts of farms on individual fish performance (e.g., through disease, pollutants, or food augmentation). In particular, we adapt the model to include these ocean farm specific characteristics to explore how ocean aquaculture farms may affect population biomass and fishery catch. We implement the model using an annual time step in a simulation area that is large enough to capture changes in parameters of interest; population and fishery outcomes are reported at equilibrium as the difference between population and fishery outcomes with and without the addition of a farm. All modeling and analyses were conducted in R(41) and all data and code are available online: github.com/couture322/oceanFarms."

Reviewer 2

Line 63-67. This is a run-on, I would split this up. Edited to break up this sentence

Line 88-92. This is another run-on. Edited to break up this sentence

Line 88-98. Maybe I missed it, but is there any evidence that fish farms act as FADs? The only papers cited here don’t specify fish farms as FADs in any way. While I generally agree with you, attraction is a major variable within the model presented. I would add at least a couple citations backing up this statement. References added that support attractive nature of different farms, including fish farms.

Line 99-108. This is my biggest discrepancy with the authors. While yes, MPAs serve as stock rebuilding functions, is there any actual evidence that fish farms specifically act as MPAs? If they do act as MPAs, is there any evidence that they are effective? Besides the fact that farms physically restrict fishing, MPAs largely function because of strategic placement. They are used to protect a large breeding population, specific vulnerable life history stages, breeding grounds, entire ecosystems etc. Is the argument that a random fish farm in the middle of the ocean does the same thing? Does this only apply to fish farms that are specifically within MPAs already? Is the spillover argument that the fish in the farm are producing gametes that are fertilized by other farmed animals or the wild populations to increase wild populations? This is unclear and not very well supported within the text. Literature cited is related to marine reserves, but nothing is cited for fish farms acting as MPAs. While this is going into the weeds a little bit, this is a foundational assumption within the model. This needs to be well cited or have data provided very clearly otherwise it undermines the conclusions based on the model. 

We agree with the reviewer that this critical point was not well explained or supported. We have substantially edited this paragraph to both clarify the theoretical nature of our exploration and highlight a key example from the peer-reviewed literature that provides evidence that real world farms can act as “MPAs”:

“When fish become resident at farms(16,33) the protection conferred by aquaculture farms could also function as an ‘MPA’, serving a stock rebuilding function, particularly in overfished or weakly regulated fisheries (19, 34,35). Such protections could potentially provide spillover benefits to surrounding fisheries for some stocks (36,37). For example, a seabream farm in the NW Mediterranean found that farms were conferring protection to commercially targeted fish stocks; however, harvest by farm employees inside the farm prevented stock rebuilding and spillover from occurring (19). Beyond protection, farms could serve to further modify wild fish population dynamics through the addition of nutrient input, e.g., via excess feeds, to the surrounding environment. Depending on the context and quantity of inputs, such added nutrients may augment or undermine these purported population and fishery benefits.”

Line 135-140. Run-on sentence, split up. Also, it needs to be specified which values for the fish life history were used (growth, reproduction, etc), and I would put them here. If they were pulled from Ovando et al., 2016, specify which species. Again, weeds, but these are important variables. We have rewritten this section to better describe the Ovando et al. model and our extension of it:

“ Briefly, the Ovando et al. 2016 model simulates fish movement and fishery behaviors around areas of protection to test the biological and economic outcomes of different management scenarios. We then expand this model to include two additional features: 1) the potential attractive impacts of farms on fish dynamics similar to the roles of FADs, and 2) the potential negative/positive impacts of farms on individual fish performance (e.g., through disease, pollutants, or food augmentation).”

Line 148. Very well-defined variables in this and the next section that make it clear what is being used in the model. Thanks!

Line 150-153. This is a major point of contention within the model. After a brief look at Ovando et al.’s 2016 paper, they used a 1-dimensional model to represent fish dispersal around an island, correct? Is this a model framework that theoretically aligns with such a broad model as described in this manuscript? This model should make the same assumptions that the original paper did, and limit the conclusive statements provided to that situation, or alter the model framework to reflect a more general situation. Our model does make the same assumptions as the Ovando et al. model; in both instances, fish movement is simulated across an area with and without an MPA (in Ovando) and a farm (here). There are no islands in the Ovando et al. model (nor in our approach). In both Ovando et al. and here, fish population dynamics (i.e., biomass) and fishing dynamics (i.e., catch) are simulated with and without the MPA (Ovando) or farm (here), and results are reported as the difference between population and fishery outcomes with and without the addition of an MPA (Ovando) and farm (here). We have edited the methods to clarify these points:

"We implement the model using an annual time step in a simulation area that is large enough to capture changes in parameters of interest; population and fishery outcomes are reported at equilibrium as the difference between population and fishery outcomes with and without the addition of a farm.”

Line 172-174. These definitions make sense, but what are the actual inputs for the model, and how do they differ? The effort parameters are stock specific, and so rather than report a value for the stock simulated here, we have included additional details to highlight that the stock specific nature of the parameters:

"Weak fishery management is represented by a high-value fishery under open access management. In this case, fishing effort is calculated and adjusted at each timestep at the level that maximizes profits. Strong fishery management is represented by a constant effort that generates maximum sustainable yield for the simulated stock.”

Line 200-202. Citation? References added

Line 206-207. This is why it is so important to provide evidence that fish farms are actively playing an MPA role and not just a FAD role. This is a major factor within the model that is largely unsupported. We agree and have revised the introduction (noted above as well) to clarify this point. 

“When fish become resident at farms(16,33) the protection conferred by aquaculture farms could also function as an ‘MPA’, serving a stock rebuilding function, particularly in overfished or weakly regulated fisheries (34,35) . Such protections could potentially provide spillover benefits to surrounding fisheries for some stocks (36,37). For example, a seabream farm in the NW Mediterranean found that farms were conferring protection to commercially targeted fish stocks; however, harvest by farm employees inside the farm prevented stock rebuilding and spillover from occurring(19). Beyond protection, farms could serve to further modify wild fish population dynamics through the addition of nutrient input, e.g. via feeds, to the surrounding environment. Depending on the context and quantity of inputs, such added nutrients may augment or undermine these purported population and fishery benefits.” (lines 102 – 112)

Line 226-228. Why 10%? Is there any reason for this number or is 10% largely arbitrary? This value is largely arbitrary. We wanted a level that was high enough to affect the population model, but not high enough to kill everything off. We have added the following sentence to the text to clarify this point: 

“The 10% change in mortality is arbitrary and was selected to be large enough to result in detectable population impacts, but not so large as to overwhelm the population dynamics and become uninterpretable.”

Line 229. I think you should add all of the figures in the supplementary information to the main text. It’s better to show the results of the model instead of just talk about them. 

Thank you for your comment. We moved one figure up to the main text that is referenced multiple times, but felt the rest were supplementary. 

Line 237-239. I think these are incredibly useful metrics being presented here. However, is even a 10% area realistic simply due to farming? These are farms that are ~1 square km, as cited in this manuscript. More evidence that the scope of this manuscript might need to be narrowed to make the results more believable. We agree with the reviewer that the realism/real world application of our simulation is not obvious. Ultimately, we decided not to be overly restrictive in modeling farm area a priori because real world farms often exist in small bays where populations may have little to no genetic connectivity to other populations. Alternatively, as is seen in locations with truly industrial scale farming operations, farms can cover enormous ocean area. Clarifying language was added to the methods:

“Farm size is modeled relative to the population’s range size and is varied from 0% to 40% to represent a range of possible scenarios including smaller farms in restricted spaces or affecting species with smaller home ranges, to large industrial farms affecting species of differing mobilities.” (lines 237 – 240)

Line 256. Sessile is not a life history pattern observed in fish. While this is a concise term within the model to indicate no movement out of the farm once settled, it does not describe fish ecology. Yes! But while these do not describe finfish species, fisheries can also target sessile species such as mussels, so we wanted to include this level of movement as well. We added a note for this example in the text: “(movement = 0, eg., mussel fisheries)”

Line 297-303. I possibly missed it, but is there any graphical representation of the biomass under positive impacts? They are discussed in this paragraph but no figure is cited. The positive impacts referred to here are built into the standard farm model (increased growth rate, lines 206-207) and so the relevant results are included in Figure 3. Negative impacts are modeled and included in the Supplementary Information: Fig S7

Line 323-340. I appreciate the author’s observations about the limitations of the model being presented. However, without providing any real-world data or evidence, why should the readers believe any of the results of this model can be applied to real-world ecosystems? We appreciate the need to connect models and their parameters to real world situations for the model results to be relevant and useful. We tried to do this by building the logic and evidence (with citations) for the key inputs of the model: like MPAs and FADs, marine species (fish and invertebrates alike) are attracted to ocean-based farms; fishing is limited within farms, to varying degrees but often substantially; and we know that farms can impact associated wild species in positive and negative ways. Our model tests the possible outcomes from these characteristics, and we acknowledge that there is plenty of room to test and improve our model with additional in-situ studies designed to help understand under which conditions different outcomes can be expected in the real world. We hope these results can motivate additional work in this nascent field. 

Line 347-350. Wait, isn’t the argument that fish farms reduce fishing pressure due to inaccessibility? Yes, but for highly mobile and migratory species, farms may be unable to provide much protection and instead merely increase the predictability of where these individuals can be found/caught. We have added language to clarify this point. 

“…farms will likely increasingly fill the FAD role of attracting fish to a known location without being able to provide much protection, which can enhance the predictability of high fishing yield locations,…”

Line 350-352. There are plenty of FADs that exist at the moment. Is there any evidence that large FADs interrupt migration patterns right now? Yes, this is a good point. We added a note about fish presence at FADs with a reference.

“Fish associations with FADs are highly variable by species, season, other species found at the FAD, and other considerations(48). Ocean farms could cause similar volitility in aggregations or more consistency or residency given regular biomass supplementation (feeds, wastes, and/or product mortality). Therefore, further empirical research is needed to characterize if and how ocean farms might cause interruptions to migrations and longer distance movement.”

Line 366. Have real species behaviors been included in the model? Or are these abstract representations of potential behaviors? A nitpick, but very important to the interpretation of results. These are abstract representations of potential behaviors based on the limited literature accounts that exist. Specific behavioral responses are as yet unknown. The text has been edited to clarify this point:

“Here we have modeled several species behaviors to mimic common conditions described at farms but there are additional behaviors that require further exploration.”

Line 375-376. I would remove this sentence or rephrase this. Would new data then be incorporated into the model presented, or should management be based solely on empirical data? This sentence has been edited to improve clarity and highlight that empirical data can be used to improve the model and output results:

“Lessons learned can then be used to better parameterize this model to better inform planning and management of regional aquaculture improve understand of potential risks to cultured and wild species.”

Line 410-412. This is why validation specifically from the authors is sorely needed. Presenting a model without validation reduces the usability of the model. We agree that empirical validation of theoretical models is a highly valuable part of research, but there is a long history of theoretical ecology presenting and testing ideas by incorporating known (or expected) assumptions about system attributes and simulating the outcomes. Our approach here is of this lineage of theoretical research.

Line 417-420. This begs the question of why commonly available parameters weren’t included in the model in the first place. Wouldn’t that provide a stronger case for using this model? We used realistic ranges of parameters for those we tested to see how they affect the results but could not fully model (with specific parameters) any one species because attraction and aggregation rates do not exist for any species. To help clarify this, we have added the following sentences:

“The results here show that, in theory, ocean farms can have positive or negative impacts on wild populations and local capture fisheries depending on several design attributes and traits of species of interest.” (lines 327 – 329)

“For the vast majority of parameter combinations, results showed that farms could have positive conservation effects on overall fish biomass and disproportionate increases in total biomass compared to catches…” (lines 329 – 331)

“Therefore, while we may see a potential for consistent benefits to total biomass from aggregation at a farm, as farms take up substantially more space these conservation benefits may ultimately come at a cost to fisheries yields if rates of aggregation are too high.” (lines 351 – 353)

“although ocean farming across most of the globe still occurs at a scale small enough that benefits could be achieved from significant future expansions and increases in population biomass can help create a buffer to management uncertainty in the face of decreased catch.” (lines 354 – 357)

“Second, our results show the potential for farms to provide strong benefits to conservation in an area by providing refuge from fishing under both weak and strong fisheries management.” (Lines 384 – 386)

“Interactions between aggregation at farms and smaller farm areas could provide synergistic benefits to local fishing, especially for smaller to moderate total farm areas” (lines 406 – 407)

“We show that the interaction between ocean farms and wild capture fisheries likely depend significantly on the design of the farm and the ecology of the target species.” (lines 438 – 439)

“We found that when used strategically, ocean-based aquaculture has the potential to benefit both wild populations and fishery catches.” (lines 472 – 473). We found that when used strategically, ocean-based aquaculture has the potential to benefit both wild populations and fishery catches. (lines 472 – 473)

Line 423-424. Case in point for why conclusions and conclusive language need to be softened within the manuscript. Thank you for this feedback, we made edits throughout to soften the messages and conclusions, as such:

“The results here show that, in theory, ocean farms can have positive or negative impacts on wild populations and local capture fisheries depending on several design attributes and traits of species of interest.” (lines 327 – 329)

“For the vast majority of parameter combinations, results showed that farms could have positive conservation effects on overall fish biomass and disproportionate increases in total biomass compared to catches…” (lines 329 – 331)

“Therefore, while we may see a potential for consistent benefits to total biomass from aggregation at a farm, as farms take up substantially more space these conservation benefits may ultimately come at a cost to fisheries yields if rates of aggregation are too high.” (lines 351 – 353)

“although ocean farming across most of the globe still occurs at a scale small enough that benefits could be achieved from significant future expansions and increases in population biomass can help create a buffer to management uncertainty in the face of decreased catch.” (lines 354 – 357)

“Second, our results show the potential for farms to provide strong benefits to conservation in an area by providing refuge from fishing under both weak and strong fisheries management.” (Lines 384 – 386)

“Interactions between aggregation at farms and smaller farm areas could provide synergistic benefits to local fishing, especially for smaller to moderate total farm areas” (lines 406 – 407)

“We show that the interaction between ocean farms and wild capture fisheries likely depend significantly on the design of the farm and the ecology of the target species.” (lines 438 – 439)

“We found that when used strategically, ocean-based aquaculture has the potential to benefit both wild populations and fishery catches.” (lines 472 – 473)

Line 429-431. The range of species here all have different very different levels of movement, dispersal, age/length at maturation, and fecundity. Won’t these need to be modeled differently to account for the disparity in all of these variables? Yes! We focused our analyses on the attractive and aggregation potential of the farms which affect movement of adults, and therefore varied this parameter over a realistic range to understand how difference will affect responses to farms and impacts on populations and fishing. We agree that larval movement and fecundity will also impact outcomes here, but we determined that these parameters were outside of the scope of this manuscript. 

Line 431-435. It’s not quite clear why these species are included here. None of these organisms, which again have very different life histories and dispersal methods, were included in the model. Won’t these species need a completely different model to get accurate results from simulated situations? These species are mentioned as a discussion of potential trends we might expect and to bring in some existing information, while also highlighting that responses will likely vary. This last point supports our decision to keep the model abstract and not model a specific species, farm, or environment. 

Line 455-457. Is there any evidence that this is true? Not all fish species are attracted to FADs. Bacher & Gordoa 2016 show that fish are attracted to fish farms and protected from commercial fishing within the lease area. While not all species are attracted to traditional FADs, farms provide additional benefits (food, nutrients) and structure, to which more species are attracted compared to traditional offshore FADs, which under several cases (artificial reefs, decommissioned oil platforms, aquaculture, etc) have proven to increase biomass and diversity in an area. 

We have added the Bacher & Gordoa 2016 reference to this line to support these ideas.

Bacher K, Gordoa A. Does marine fish farming affect local small-scale fishery catches? A case study in the NW Mediterranean Sea. Aquaculture Research. 2016 Aug;47(8):2444–54.

---

## [Decision Letter · Decision Letter 1]

25 Jan 2024

Could fish aggregation at ocean aquaculture augment wild populations and local fisheries?

PONE-D-23-28349R1

Dear Dr. Couture,

We’re pleased to inform you that your manuscript has been judged scientifically suitable for publication and will be formally accepted for publication once it meets all outstanding technical requirements.

Kind regards,

José A. Fernández Robledo, Ph.D.

Academic Editor

PLOS ONE

Additional Editor Comments (optional):

Reviewers' comments:

Reviewer's Responses to Questions

**Comments to the Author**

1. If the authors have adequately addressed your comments raised in a previous round of review and you feel that this manuscript is now acceptable for publication, you may indicate that here to bypass the “Comments to the Author” section, enter your conflict of interest statement in the “Confidential to Editor” section, and submit your "Accept" recommendation.

Reviewer #2: All comments have been addressed

2. Is the manuscript technically sound, and do the data support the conclusions?

Reviewer #2: Yes

3. Has the statistical analysis been performed appropriately and rigorously? 

Reviewer #2: Yes

4. Have the authors made all data underlying the findings in their manuscript fully available?

Reviewer #2: Yes

5. Is the manuscript presented in an intelligible fashion and written in standard English?

Reviewer #2: Yes

6. Review Comments to the Author

Reviewer #2: I want to thank the authors for addressing all of my concerns succinctly and clearly. The reduction of hard conclusions has done a lot to present this model as purely theoretical. I recognize the difficulty in testing these results in real-world situations and thus the need for theoretical modeling to understand the potential effects of these management decisions. The manuscript, as is, presents potentially testable hypotheses for future work to explore. However, without the presentation of a case study, I do wonder if the broad nature of this model will prevent it back from proper predictions of real-world ecology. With that said the authors have done enough to properly answer my concerns and preface their previously hard conclusions with proper ambiguity. I can recommend this manuscript for publication.

7. PLOS authors have the option to publish the peer review history of their article (what does this mean?). If published, this will include your full peer review and any attached files.

Reviewer #2: **Yes: **Patrick T. Rex

---

## [Editor Report · Acceptance letter]

27 Feb 2024

PONE-D-23-28349R1 

PLOS ONE

Dear Dr. Couture, 

I'm pleased to inform you that your manuscript has been deemed suitable for publication in PLOS ONE. Congratulations! Your manuscript is now being handed over to our production team.

Kind regards, 

on behalf of

Dr. José A. Fernández Robledo 

Academic Editor

PLOS ONE